# Variation in thermal physiology can drive the temperature-dependence of microbial community richness

Tom Clegg[1,2,3]*, Samraat Pawar[3]

[1]Helmholtz Institute for Functional Marine Biodiversity at the University of Oldenburg (HIFMB), Oldenburg, Germany; [2]Alfred Wegener Institute, Helmholtz Centre for Polar and Marine Research (AWI), Bremerhaven, Germany; [3]Department of Life Sciences, Silwood Park Campus, Imperial College London, Ascot, United Kingdom

**Abstract** Predicting how species diversity changes along environmental gradients is an enduring problem in ecology. In microbes, current theories tend to invoke energy availability and enzyme kinetics as the main drivers of temperature-richness relationships. Here, we derive a general empirically-grounded theory that can explain this phenomenon by linking microbial species richness in competitive communities to variation in the temperature-dependence of their interaction and growth rates. Specifically, the shape of the microbial community temperature-richness relationship depends on how rapidly the strength of effective competition between species pairs changes with temperature relative to the variance of their growth rates. Furthermore, it predicts that a thermal specialist-generalist tradeoff in growth rates alters coexistence by shifting this balance, causing richness to peak at relatively higher temperatures. Finally, we show that the observed patterns of variation in thermal performance curves of metabolic traits across extant bacterial taxa is indeed sufficient to generate the variety of community-level temperature-richness responses observed in the real world. Our results provide a new and general mechanism that can help explain temperature-diversity gradients in microbial communities, and provide a quantitative framework for interlinking variation in the thermal physiology of microbial species to their community-level diversity.

*For correspondence: thomas.clegg@hifmb.de

Competing interest: The authors declare that no competing interests exist.

## Editor's evaluation

This important study proposes a phenomenologically motivated theoretical framework to explain observed patterns of the temperature dependence of microbial diversity. The methodology is overall convincing. The manuscript should be of interest to microbial ecologists.

## Introduction

The effect of temperature on biodiversity has long been a topic of interest in ecology. Starting with the pioneering work of Alexander von Humboldt, who in the 19th century identified temperature as a major environmental driver of plant richness along elevational gradients in the Andes (*Humboldt and Bonpland, 2010*), temperature has been recognized as a key driver of the geographical gradients in taxonomic richness are seen across practically all organismal groups (*Rohde, 1992*; *Gaston, 2000*). In recent years, the relationship between species richness in microbial communities and temperature has become a topic of particular interest. This has come together with an increase in awareness of the importance of these microbes to ecosystem functioning (*Schimel, 2013*; *Graham et al., 2016*; *Antwis et al., 2017*), and new DNA sequencing technologies that allow community 'snapshots' to be characterised with relative ease (*Zimmerman et al., 2014*). Studies on microbial community richness,

often measured in numbers of OTUs (operational taxonomic units), have generally found varying responses to changes in environmental temperature. For example, while *Zhou et al., 2016* found that soil microbe richness increased across a continental temperature gradient in North America, others have found unimodal responses (richness peaking at intermediate temperatures) in soils as well as other environments (*Milici et al., 2016*; *Sharp et al., 2014*; *Thompson et al., 2017*). Indeed, as demonstrated in the data synthesis by *Hendershot et al., 2017*, the temperature responses of microbial richness or diversity are 'consistently inconsistent,' with no single pattern in terms of shape (monotonic or unimodal) or direction (positive or negative) dominating.

Currently, there are two mechanistic explanations relevant to microbial temperature-richness gradients, both of which focus on energy availability in the environment. The first is the metabolic theory of biodiversity (MTB) (*Allen et al., 2002*), which predicts monotonic increases in species richness with temperature due to increasing cellular kinetic energy at higher temperatures. This allows more individuals to survive in a given community, which in turn supports higher species richness. This work was later extended by *Arroyo et al., 2022* who were able to produce a variety of temperature-diversity responses by including a more complex model of enzyme kinetics allowing for unimodal responses. The MTB and its newer applications are able to recreate various temperature-diversity patterns but rely on three key assumptions: (1) all populations have the same rate of energy use (energy equivalence), (2) identical temperature dependence across taxa (the 'Universal Temperature Dependence' or 'UTD'), and (3) non-interacting populations. Whilst there is some evidence for the energy equivalence in phytoplankton communities (*Ghedini et al., 2020*) its validity remains, to the best of our knowledge, untested in heterotrophic microbes. Support for the other assumptions is weaker and there is now extensive evidence for significant functional variation in thermal sensitivities across the microbial tree of life (*Smith et al., 2019*; *Dell et al., 2011*; *Kontopoulos et al., 2020*) emphasising the fact that the UTD is at best an approximation (*Savage, 2004*). Likewise, extensive theoretical and empirical evidence shows that resource-mediated species interactions among microbes are the norm and drive community species dynamics and diversity (*Goldford et al., 2018*; *Marsland et al., 2019*; *Ratzke et al., 2020*; *Cook et al., 2021*; *Lechón et al., 2021*).

A second explanation for temperature-diversity gradients is the metabolic niche hypothesis (*Sharp et al., 2014*) which posits that there are more energetically viable ways to make a living at intermediate (non-extreme) temperatures. This allows for species coexistence and in turn produces an unimodal temperature-diversity response (*Clarke and Gaston, 2006*). This mechanism was modelled phenomenologically by *Marsland et al., 2020* who imposed additional mortality on consumers to represent less-favorable environmental conditions, recovering the expected unimodal patterns of richness. Overall the metabolic niche hypothesis assumes that the size of the feasible niche space follows a specific pattern over temperature and is thus unable to explain other richness-temperature relationships. Likewise, it is unable to explain how these effects arise explicitly from the action of temperature on individual populations and their thermal responses.

A key weakness in these current explanations is the UTD assumption; that focusing on the average of thermal responses is an appropriate approximation (*Savage, 2004*). We posit that the variation in thermal responses will in fact be important in determining the responses of microbial community richness to temperature. In addition to its ubiquity (*Smith et al., 2019*; *Smith et al., 2021*; *Kontopoulos et al., 2020*) variations on thermal responses may act in two ways. First, the nonlinear thermal responses of metabolic traits means that inter-specific variation in thermal sensitivity will likely drive significant changes in realised trait-value distributions and species interactions at different temperatures. Second, differences in thermal responses of traits between interacting populations ('physiological mismatches') may have non-trivial effects on microbial community dynamics and coexistence (*Dell et al., 2014*; *Bestion et al., 2018*; *García et al., 2023*).

In this paper, we derive a new theory that predicts the response of species richness of microbial communities to temperature while accounting for variation in thermal sensitivity of metabolic traits across populations. We focus on competitive interactions which have been shown to have string effects on coexistence and richness in microbial communities (*Marsland et al., 2019*; *Goldford et al., 2018*; *Ratzke et al., 2020*; *Lechón et al., 2021*). We first derive a mathematical expression that links the distribution of population thermal performance curves to the number of species that can feasibly coexist within a community. Then, using empirical data to parameterise the model, we ask whether

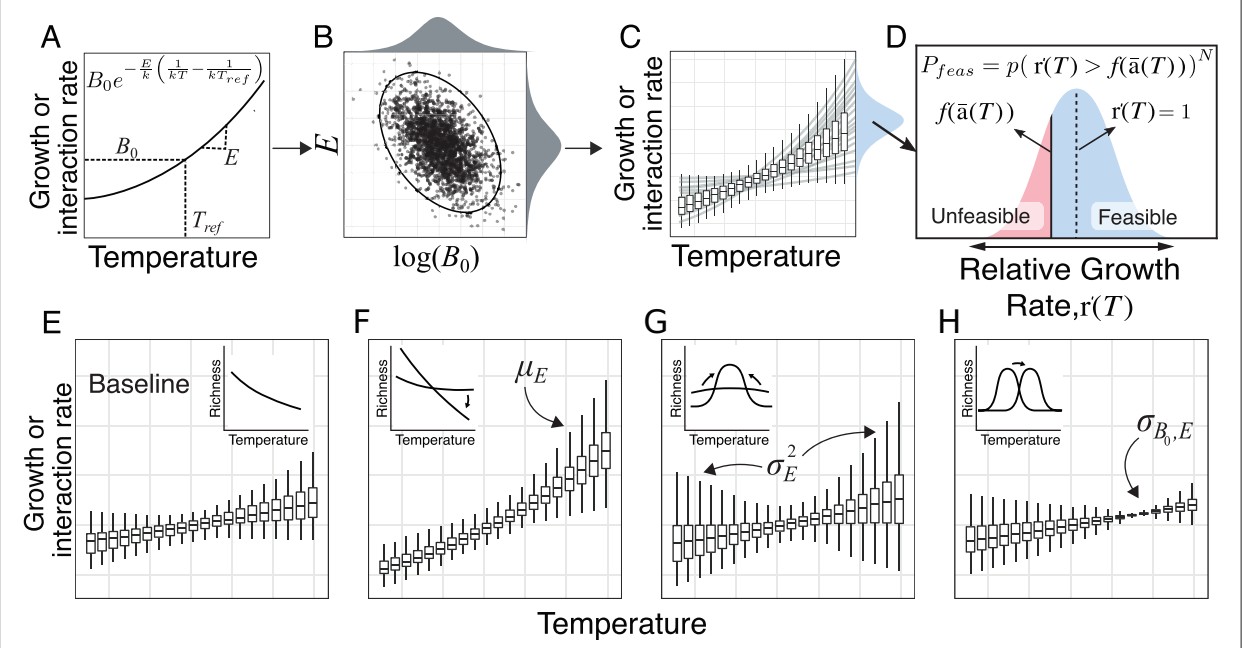

**Figure 1.** How variation in thermal physiology constrains microbial community species richness. (**A**) Trait values increase with temperature following the Boltzmann-Arrhenius equation (boltz_maintext), with the shape governed by two parameters: $B_0$ - trait value ($r$ or $a$) at a reference temperature $T_{ref}$ and $E$ - thermal sensitivity. (**B**) The joint distribution of $E$ and $\log(B_0)$ (here with empirically realistic negative covariance) determines how trait distributions vary across temperatures (**C**). (**D**) The distribution of trait values in turn determines the probability of feasibility $P_{feas}$ (and thus richness; Feas_sp_maintext). Specifically, $P_{feas}$ is determined by the proportion of relative growth rates ($r'_i$; blue shaded area) that are greater than the bound (solid black line) set by mean interaction strength ($\langle a \rangle$). Populations with relative growth rates below this bound (red shaded area) are unfeasible (cannot persist in the community). All else being equal, the size of the unfeasible region (i.e. richness), decreases with increasing variance in the growth rate distribution ($\mathrm{Var}(r'_i)$) and increasing interaction strength (which shifts the $f(\langle a \rangle (T))$ bound upwards). (**E–H**) The effects of varying different aspects of the joint distribution of $B_0$ and $E$ of $r$ and $a$ on the emergent trait distribution across temperatures. Each panel shows the effect of altering the labeled parameter relative to the baseline case (far left), with inset plots showing the effect on the resulting temperature-richness relationship.

the extant variation in thermal responses of bacterial metabolic traits is sufficient to be a key drivers of patterns of species richness across temperature gradients in the real world.

## Results

### Theory

In order to investigate the effects of temperature on community richness we first link the effects of the community-level distributions of two key traits—maximal population growth rate $r_i(T)$ and pairwise interaction strengths $a_{ij}(T)$ —to the probability of feasibility ($P_{feas}$): the probability that the community will support all species' populations at non-zero abundance at equilibrium. Feasibility is a necessary condition for stable population coexistence and generally falls as richness increases, placing an upper bound on community size (**Goh and Jennings, 1977**; **Grilli et al., 2017**; **Dougoud et al., 2018**). We then determine how temperature, acting through its effect on metabolic rate, affects the distributions of traits across the community, accounting for the variation in thermal responses across species in the community. Finally, we combine these to determine the effect of temperature on feasibility and thus the maximal richness. **Figure 1** provides an overview of the theory.

### Community-level trait distributions determine species richness

In order to determine the maximal community richness we start with the generalised Lotka-Volterra model (GLV) which describes the population dynamics of a $N$ species community

$$\frac{1}{x_i}\frac{dx_i}{dt} = r_i(T) - a_{ii}(T)x_i - \sum_{j=0}^{N} a_{ij}(T)x_j \tag{1}$$

where $x_i$ is the biomass of the species $i$, $r_i(T)$ is its mass-specific growth rate, and $a_{ij}(T)$ and $a_{ii}(T)$ are the inter- and intraspecific interaction strengths between and within populations. Note these parameters are expressed as functions of temperature $T$, the form of which will be discussed later.

Using a mean-field approximation (**Wilson et al., 2003**; **Rossberg, 2013**) we derive a condition for community feasibility which depends on the distributions of the parameters $r$, $a_{ij}$, and $a_{ii}$ across the community. This approximation assumes that the community we consider is large such that interactions can be considered in terms of their average value and the effect of any individual interaction is small. We discuss these assumptions in more detail in SEC:Methods. The approximation lets us write an expression for the probability that a community of a given size is feasible $P_{feas}$ as

$$P_{feas} = P\left(r_i'(T) > \frac{(N-1)\langle a\rangle(T)}{1+(N-1)\langle a\rangle(T)}\right)^N. \tag{2}$$

Here $r_i'(T) = r_i(T)/\langle r_i\rangle(T)$ is the normalised growth rate (i.e. growth rate relative to the average of all $N$ populations), and $\langle a\rangle(T) = \langle a_{ij}\rangle(T)\langle a_{ii}^{-1}\rangle(T)$ is the effective interspecific interaction strength (normalised by intraspecific interactions $\langle a_{ii}^{-1}\rangle(T)$). The inequality inside the brackets represents the probability that a given population is feasible (i.e. has non-zero biomass) with the $N$th power term representing the fact that all populations must meet this criteria for a community to be feasible.

Feas_sp_maintext shows that community feasibility changes with system size in two ways. First, assuming that the average strength of individual competitive interactions is constant, the addition of new species to a community will result in the overall strength of competition increasing ($(N-1)\langle a\rangle(T)$). This reduces the chance that the inequality in Feas_sp_maintext holds and each individual population is feasible. Second, the inequality must hold across all $N$ species, the probability of which falls as system size increases, reducing $P_{feas}$. Together, these two mechanisms place an upper limit on the size of a community that is likely to remain feasible. This limit can be calculated by setting an threshold value for $P_{feas}$ and then solving Feas_sp_maintext for $N$ (see below).

## Variation in thermal physiology determines temperature-specific trait distributions

Having derived the condition for feasibility and the limit it places on richness, we now consider how the distributions of and variation in growth rate $r$ and interaction strengths $a_{ij}$ and $a_{ii}$, change with temperature, and how this, in turn, affects the richness of species through $P_{feas}$. We use the Boltzmann-Arrhenius equation which describes the change in a given trait over temperature (**Figure 1A**; **Gillooly et al., 2001**):

$$B(T) = B_0 e^{-\frac{E}{k}\left(\frac{1}{kT}-\frac{1}{kT_{ref}}\right)} \tag{3}$$

where $B(T)$ is the trait value, $T$ is the temperature in Kelvin, $B_0$ is the normalisation constant which defines the trait value at some chosen reference temperature $T_{ref}$, $E$ (eV) is the thermal sensitivity which determines the change in trait value to a unit change of $1/kT$, and $k$ is the Boltzmann constant.

The Boltzmann-Arrhenius equation is a sufficiently accurate model for the temperature dependence of metabolically 'higher-level' traits such as interaction and growth rates, because these ultimately emerge from cellular biochemical kinetics (**Gillooly et al., 2001**; **Savage, 2004**; **Dell et al., 2011**; **Dell et al., 2014**; **Arroyo et al., 2022**) (see SEC:Methods). The empirical validity of Boltzmann for $r$ is now well-established (**Smith et al., 2019**; **Kontopoulos et al., 2020**). In contrast, there is currently no empirical evidence that directly supports its validity for the temperature dependence of interaction strengths $a_{ij}$ and $a_{ii}$ for heterotrophic microbes. We posit that a Boltzmann-Arrhenius (or at least exponential-like) temperature dependence of interaction strength is likely to be a good description, however, because pairwise microbial competitive interactions are ultimately driven by the two species' resource uptake rates, as shown by the derivation of effective interaction strengths in more mechanistic consumer-resource models of microbial communities (**Marsland et al., 2020**). As uptake rates are known to follow a Boltzmann-Arrhenius form within the OTR (**Smith et al., 2021**; **Bestion et al., 2018**) it follows that the interaction strength may follow this exponential-like form too. Finally, we note that we implicitly assume that variation in growth and interaction rates stem from cellular

metabolic processes unlimited by resource supply (*Savage et al., 2004*), i.e., we are assuming here that resource supply is sufficient to maintain positive growth rates across the community.

To derive an expression for the temperature-dependent distribution of traits we consider how $E$ and the logarithm of $B_0$ vary across the community. We assume these follow a bivariate normal distribution parameterised by the means $\mu_{B_0}$ and $\mu_E$, variances $\sigma^2_{B_0}$ and $\sigma^2_E$, and covariance $\sigma_{B_0,E}$ (theoryB). A bivariate normal distribution captures the mean and variance of the thermal dependence of these traits across the community, as well as the covariance between them. This covariance is important and generally expected to be negative due to the well-known thermal specialist-generalist trade-off (*Huey and Hertz, 1984*; *Angilletta, 2009*; *Kontopoulos et al., 2020*) that individuals cannot perform equally well at all temperatures; as a result, they can either increase performance across a narrow range of temperatures (specialist with high sensitivity $E$ but low-performance $B_0$) or perform at a lower level across a wider range of temperatures (low $E$, high $B_0$). Applying Boltzmann to these traits yields an expression for $B(T)$ that follows a log-normal distribution:

$$\log(B(T)) \sim \mathcal{N}\left(\mu_B(T), \sigma^2_B(T)\right) \quad \text{where} \quad \begin{aligned} \mu_B(T) &= \mu_{B_0} - \mu_E \left(\frac{1}{kT} - \frac{1}{kT_{ref}}\right) \\ \sigma^2_B(T) &= \sigma^2_{B_0} + \sigma^2_E \left(\frac{1}{kT} - \frac{1}{kT_{ref}}\right)^2 - 2\sigma_{B_0,E}\left(\frac{1}{kT} - \frac{1}{kT_{ref}}\right). \end{aligned}$$

(4)

It is important to note that because $B(T)$ is log-normally distributed, its moments depend on both the underlying mean and variance, $\mu_B(T)$ and $\sigma_B(T)$, respectively. boltz_maintext reveals three key insights into the effects of temperature on the distributions of the two key traits:

1. A higher mean thermal sensitivity ($\mu_E$) across species in the community increases not just the mean trait value with temperature but also its variance (*Figure 1F*).
2. Increasing variance in thermal sensitivity ($\sigma^2_E$) increases trait variance at extreme temperatures (indicated by the quadratic temperature term). In the absence of covariance, this occurs either side of the reference temperature $T_{ref}$ (*Figure 1G*).
3. The covariance $\sigma_{B_0,E}$ determines the temperature where the lowest trait variance occurs because of the linear temperature term. Negative covariance (as expected from the thermal specialist-generalist trade-off) shifts this point towards warmer temperatures (*Figure 1H*).

We can also derive a condition for the point at which this variation is sufficient to induce a unimodal response in the mean trait value, $\sigma^2_E > \mu_E + \sigma_{B_0,E}$, that is the variation in thermal sensitivity must be larger than its average over the community plus the effects of covariance. As the covariance is expected to be negative this relaxes the bound, and reduce the degree of variation needed for a unimodal response.

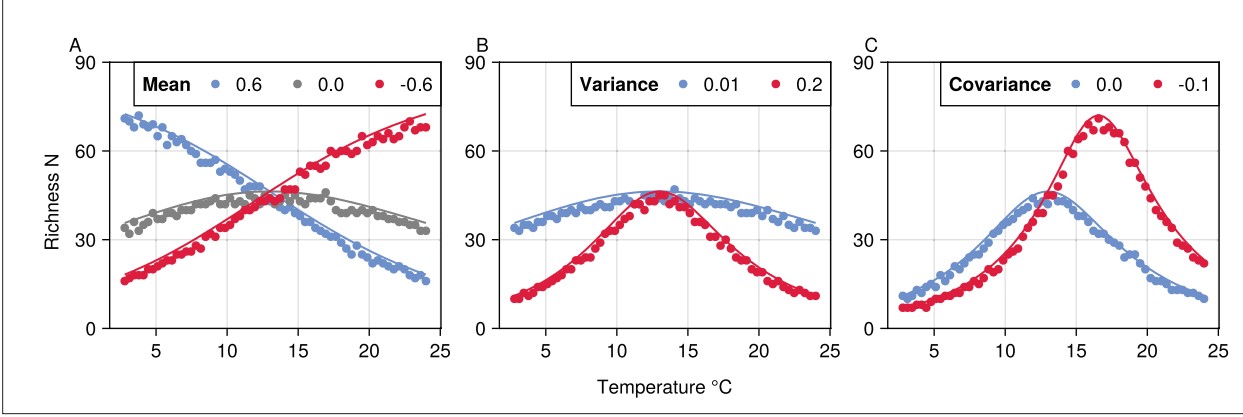

**Figure 2.** The effect of variation in trait thermal performance curves (TPCs) on the temperature-richness relationship in competitive microbial communities. The analytical predictions (solid lines) are plotted along with the maximum richness reached in the numerical simulations (dots). (**A**) Mean thermal sensitivity of interactions $\langle a \rangle$ determines the direction and steepness of the temperature-richness relationship. (**B**) Increasing variance of thermal sensitivity increases unimodality. (**C**) Negative covariance between $B_0$ and $E$ shifts the peak of richness to higher temperatures. Parameter values used were: $\mu_{r0} = 0.0, \sigma^2_{r0} = \sigma^2_{a0} = 0.2, \mu_{E_r} = \mu_{E_a} = 0.6, \sigma^2_{E_r} = \sigma^2_{E_a} = 0.01, \sigma_{B_0,E_r} = \sigma_{B_0,E_a} = 0.0$.

## Temperature determines richness by altering community-level trait distributions

With the expression for the temperature distribution of traits in hand, we now apply boltz_maintext to the two traits that determine feasibility (and thus richness) $\langle a \rangle$ and $r_i'$ (see SEC:Methods for full derivation):

$$\log \left( r_i'(T) \right) \sim \mathcal{N} \left( -\frac{\sigma_r(T)^2}{2}, \sigma_r(T) \right) \text{ and} \tag{5}$$

$$\langle a \rangle (T) = \exp \left( \mu_{aij}(T) - \mu_{aii}(T) + \frac{\sigma_{aij}(T)^2 + \sigma_{aii}(T)^2}{2} \right). \tag{6}$$

*Equations 5 and 6* show how the distribution of relative growth rate $r_i'$ at a given temperature is determined solely by the variance in $r$, while mean competitive interaction strength $\langle a \rangle$ is determined by both the mean and variance of inter- and intraspecific interaction strength $a_{ij}$ and $a_{ii}$.

By substituting *Equation 5* and *Equation 6* into the feasibility condition *Equation 2*, we can now predict the temperature-richness relationship in terms of the distributions of thermal physiology traits across species in the community *Figure 2*. This leads to three key insights.

1. The average thermal sensitivity $\mu_E$ will determine the rate at which richness exponentially changes with temperature (*Figure 1E* 2nd panel, *Figure 2A*). The response of mean effective competition $\langle a \rangle$ to temperature is determined primarily by the difference between the average thermal sensitivity of inter- and intraspecific interactions ($E_{a_{ij}}$ - $E_{a_{ii}}$) which we assume will both have a positive temperature dependence. If interspecific interactions are more sensitive ($E_{a_{ij}} > E_{a_{ii}}$) then $\langle a \rangle$ will increase with temperature resulting in the co-existence of fewer populations and lower richness. If intraspecific interactions are more sensitive ($E_{a_{ij}} < E_{a_{ii}}$) then the effective strength of competition will decrease with temperature thus leading to more populations coexisting. Note that in the case where they have the same (or no) temperature dependence the strength of effective competition will be constant over temperature and richness will be determined entirely by $r_i'(T)$.
2. Increasing variance in thermal sensitivity $\sigma_E^2$ will result in increased unimodality and a more pronounced peak in the thermal response of richness (*Figure 2B*). This effect will be primarily be determined by the variation in the thermal response of growth $\sigma_{E,r}^2$. The peak occurs because increasing $\sigma_E^2$ results in a larger variance in $r_i'$ at extreme temperatures, which means that relatively fewer species are able to endure the negative effects of competition, reducing maximum richness.
3. Negative covariance between $B_0$ and $E$ (indicative of a thermal generalist-specialist tradeoff) will shift the peak in thermal response of richness towards higher temperatures (*Figure 2C*). This happens as it shifts the point of lowest variance in growth rates to higher temperatures.

In order to visualise and test the predictions arising from Feas_sp_maintext we compared species richness patterns and the effects of changing the various thermal physiology parameters to numerical simulations using the full GLV model. To generate predictions, we selected reasonable values for thermal physiology parameters of growth rates $r$ and interaction terms $a_{ij}$ and $a_{ii}$ and substituted them into Trait_distributions_r and Trait_distributions_a. We then substituted the relevant quantities into Feas_sp_maintext across multiple temperatures and calculated $P_{feas}$ across multiple values of $N$. Then, setting a threshold value of $P_{feas} = 0.5$ (with no loss of generality) we find the maximum $N$ value a community can reach and remain above this value. To test these with numerical simulations we took the same thermal physiology parameters and generated 50 replicate communities across a temperature range with varying system sizes (sampling $r$, $a_{ij}$ and $a_{ii}$ from distributions as described by boltz_maintext). We then solved for the steady state of these communities (using the matrix form solution $x^* = A^{-1}r$) and determined which were feasible (i.e. those with no extinctions). As with the predictions we then calculated the maximum richness by calculating the $P_{feas}$ values (the proportion of replicate communities that were feasible) and selecting the largest community above or at the 0.5 threshold. *Figure 2* shows that the analytical predictions match the simulated results well and that the changes in richness over temperature respond to changes in the thermal physiology parameters as expected.

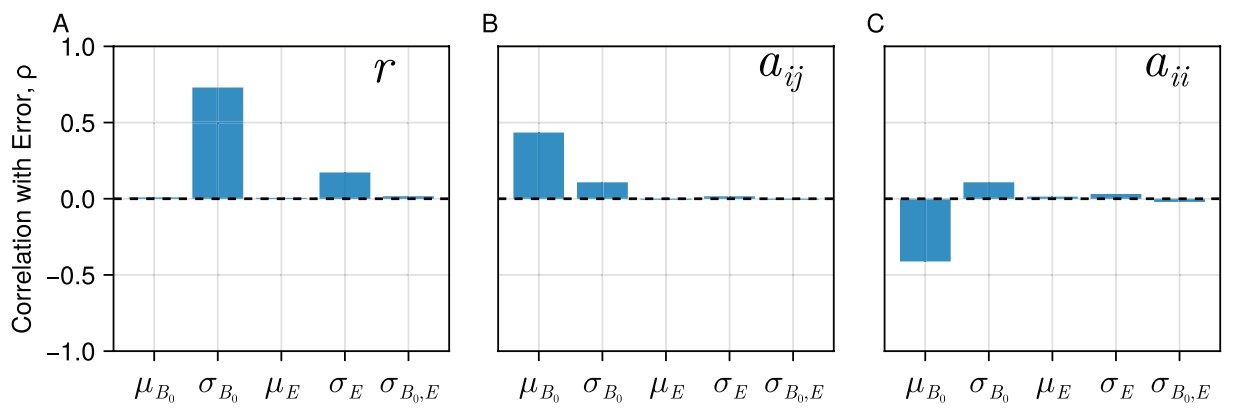

**Figure 3.** Sensitivity of theoretical results. Bar plots show the Pearson correlation coefficient $\rho = (x, y)/\sigma_x \sigma_y$ of each thermal physiology parameter with the root mean squared error between the theory and numerical simulations across replicate communities. Positive correlations indicate that increasing the parameter value tends to increase the error whilst negative values indicate error decreases the parameter value increases. Each panel shows the effect of a given parameter.

## Sensitivity analysis

We also performed a sensitivity analysis to determine the conditions under which the predictions of the theory break down. In general, we expect this will occur when the assumptions of the mean field approximation are not met, primarily when interactions are strong or their variation is large and the coupling between individual populations dominates dynamics (see SEC:Methods for more detail). To test the sensitivity of the results we generated 10,000 random communities with means and variances of the various thermal physiology parameters randomly sampled from reasonable ranges. For each community we generated the predicted and observed richness as above and then calculated the root mean squared error, the square root of the average squared difference of predicted vs simulated diversity. We normalised this error by dividing by the average richness observed (to avoid biasing the estimates with system size) and then calculated the Pearson correlation coefficient of each parameter value with the error. The choice of the measure of error or correlation is in principle not important and one could use other metrics such as $R^2$ instead. This method provides an efficient and concise way to evaluate the performance of our model and summarise the relative effect of different parameters. For a given parameter positive correlation values indicate that increasing its value leads to higher error, reducing the ability of the model to match the simulated data. conversely, a negative correlation indicates that the model performs better when the parameter is large. Overall the results are in agreement with the expectations *Figure 3*. Increasing variation in trait values $\sigma_{B_0}$ leads to increasing error in all three traits. Likewise high average strength of interspecific interactions increases error whilst increasing the average strength of intraspecific interactions decreases error.

## Real-world variation in thermal physiology predicts unimodal bacterial temperature-richness relationships

We next parameterised our model with empirical data on bacterial traits to determine the temperature-richness relationship predicted under realistic levels of variation in thermal physiology *Figure 4*. We used data on bacterial growth rates from two sources: an experimental dataset in which the growth rates of 27-soil bacteria strains were measured across a range of temperatures (*Smith et al., 2021*) and, a literature-synthesised dataset which was constructed by digitsising existing data on prokaryotic growth across 482 strains (*Smith et al., 2019*). We refer the reader to the respective papers for more details on how these data were collected. For each dataset, we refit TPCs to obtain estimates for the joint distribution of $B_0$ and $E$. Both datasets showed considerable variation in TPCs thorough variation in both $B_0$ and $E$ and a negative covariance between $\log(B_0)$ (for a $T_{ref} = 13°$ C) and $E$ values (*Figure 4A-D*). Fits to the multivariate-normal distribution using MLE yielded estimates of $\mu_E = 1.0$, $\sigma^2_{B_0} = 0.95$, $\sigma^2_E = 0.25$, and $\sigma_{B_0,E} = -0.42$ for the experimental dataset and $\mu_E = 0.82$, $\sigma^2_{B_0} = 1.0$, $\sigma^2_E = 0.11$, and $\sigma_{B_0,E} = -0.1$ for the data-synthesis. Parameterising our theory with these values (using the same thermal response for growth rates and interactions) predicts unimodal temperature-richness

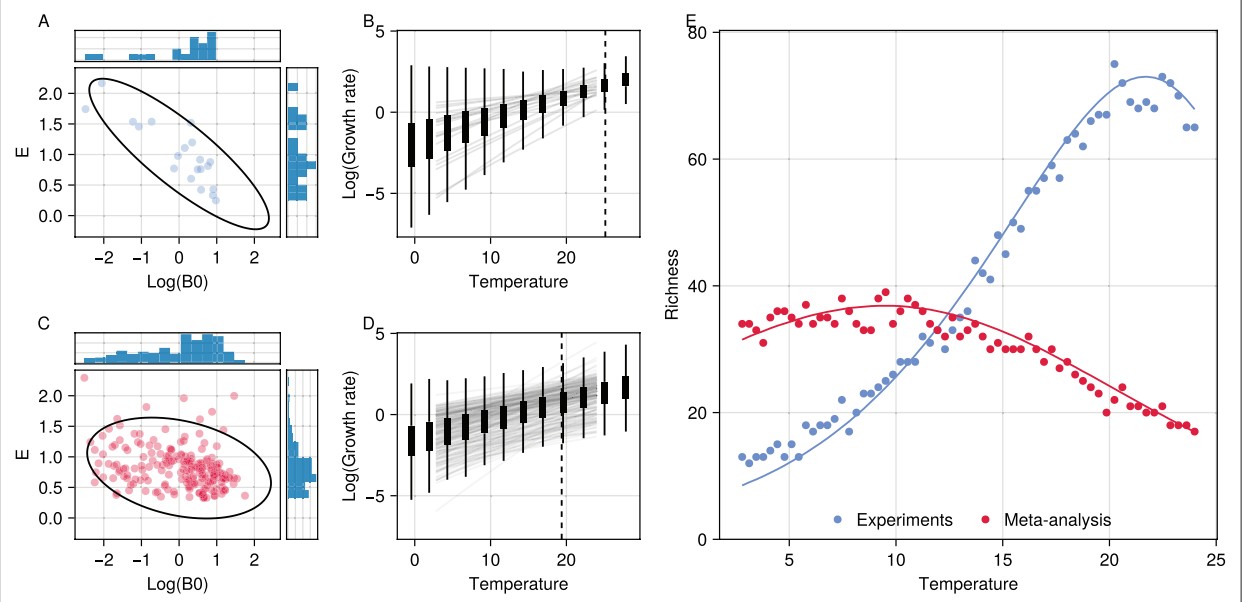

**Figure 4.** The bacterial temperature-richness relationship predicted by empirically-observed variation in thermal physiology. (**A**) The relationship between $\log(B_0)$ and $E$ for growth rate in the experimental thermal performance curve (TPC) data from *Smith et al., 2021*. Dots show each pair of $B_0$ and $E$ values estimated for a given species/strain with histograms showing the marginal distributions. Ellipses show the 95% quantiles of the fitted bivariate normal distribution. (**B**) The actual growth-rate TPCs (solid lines) from the dataset as well as the fitted trait-distributions across temperature (box plots). The dashed line shows the point of minimum variance in growth rates which occurs towards the upper end of the temperature range. (**C–D**) Analogous plots for the dataset from the literature synthesis (*Smith et al., 2019*). (**E**) The analytically- (solid line) and simulation- (points) predicted temperature-richness curves based on the TPC variation seen in both these experimental (blue) and literature-synthesised (red) empirical data. Both are generated using the parameters from their respective fitted distributions and mean normalisation constants of $\mu_{r_0} = 0.0$ and $\mu_{a_0} = -5.0$. We set the normalisation constants such that the magnitude of richness values is not too large to perform the numerical simulations.

responses due to this combination of variance and negative covariance *Figure 4E*. Due to its larger variance in $E$ as well as stronger negative covariance, the response based on the experimental data shows a sharper increase in richness, and peaks at a higher temperature of ~20°C, than that based on the data-synthesis which has a shallower, broader temperature-richness curve peaking at ~9°C.

## Discussion

We have investigated how variation in species-level thermal responses (TPCs) affects the temperature dependence of species richness in microbial communities. We show how the shapes of the across-species distributions of thermal sensitivity ($E$), the normalisation constants ($B_0$), and their covariance can determine changes in species richness over temperature. These patterns emerge as the relative strength of competition and variation in population growth rates change with temperature and can be linked directly to specific features of the thermal performance trait distributions.

A key new insight from our theory is that variance in thermal sensitivity of growth rate, $\sigma_{E,r}$, can drive unimodal patterns of temperature-richness curve (*Figure 2*). This is due to the non-linear temperature dependence of trait variance (*Equation 4*) and its effects on the community-level traits that determine richness (*Equations 5 and 6*). Furthermore, the temperature at which richness peaks is governed by the covariance between the thermal sensitivity ($E_r$) and baseline value ($r_0$) of growth rate, with negative covariance values shifting peak richness towards higher temperatures. This negative covariance case is consistent with a thermal generalist-specialist trade-off seen in existing data analysed here; *Smith et al., 2019*; *Smith et al., 2021* and suggests richness should peak towards the higher end of the operational temperature ranges (OTRs) of most mesophilic bacteria. We expect the variance and covariance of thermal response traits to play a key role in determining patterns of richness due to the extensive variation in the thermal sensitivity $E$ of metabolic traits across the microbial tree of life, as well as negative covariance between this parameter and the normalisation constant ($B_0$) (*Kontopoulos et al., 2020*; *Smith et al., 2019*).

The mechanism we present here provides an alternate explanation for the existence of temperature-diversity patterns and is based on ecological processes (i.e. competition). This represents a new type of mechanism compared to previous explanations invoking energy availability, such as the use of enzyme kinetics in the MTB (*Arroyo et al., 2022*) or the reduction in feasible niche space in the metabolic niche hypothesis (*Clarke and Gaston, 2006*). Furthermore, our model is able to produce richness peaks below the thermal optima of the underlying rates unlike the previous explanations which assume declines in richness happen due to a reduction in performance at the population level. This peak of richness below the thermal optima of individual population rates is consistent with observations of unimodal temperature-richness relationships (*Milici et al., 2016*; *Sharp et al., 2014*; *Thompson et al., 2017*) which tend to be below estimates of microbial thermal optima (*Smith et al., 2019*). Crucially we would like to stress that these mechanisms are not mutually exclusive and that the patterns of diversity observed in nature are likely the product of multiple processes acting in unison.

Overall we expect that the mechanism we propose here will be particularly relevant to predicting the temperature-richness relationship in: (i) communities where system dynamics are driven primarily by species interactions (as opposed to scenarios where dynamic assembly and processes such as environmental filtering or neutral processes dominate); (ii) environments where species typically experience temperatures within their OTR (arguably the most common scenario on planet Earth); (iii) At scales where trait TPC distributions are relatively constant across communities and thus independent of the local environment. At larger scales, we expect that processes such as local adaptation are likely to alter these distributions (*Kontopoulos et al., 2018*) as organisms adapt to local temperature regimes. More work is required to test this more explicitly, however, and will require datasets explicitly measuring the within-community variation of thermal responses across taxa.

We found that the data from the single lab experiment (*Smith et al., 2021*) show a greater variance in $E_r$ as well as a stronger covariance between $B_{0,r}$ and $E_r$ than the literature-synthesised (*Smith et al., 2019*) data (*Figure 4*). This drives a constriction of growth rate variation at ~23°C in the experimental data, which in turn results in a higher predicted peak of species richness at ~23°C these data. Estimates for $E_r$ and $B_{0,r}$ in both datasets were obtained using comparable methods, so this difference most likely reflects biological and experimental differences between them. Given that the single experimental dataset is for a far more restricted set of thermal taxa from a specific habitat (soil), it is surprising that the TPCs vary more that single community than across the wider diversity of taxa in the literature-synthesised dataset. This either reflects some sort of systematic bias in the literature data, that the local community sampled in the single experiment is a non-random set of co-evolved taxa, or both. In particular, the temperature at which the growth rate variation constricts in the lab dataset is almost identical to the temperature at which those strains were maintained, suggesting a role of species sorting, acclimation or evolution. The literature-synthesised dataset on the other hand represents a much more random set of taxa. Interestingly, the predicted ~9°C peak in species richness based on these data is almost identical to that observed by *Thompson et al., 2017* from a wide range of environmental samples, which also presumably emerges from a heterogeneous set of taxa.

In our model we use feasibility as the main constraint on species richness. We argue that feasibility is an important limit as only feasible fixed points allow the coexistence of populations within the community. Feasibility has long been discussed in the literature in this way, going back to *Goh and Jennings, 1977* who showed the scaling of system size with feasibility in GLV communities based on random parameterisations. In contrast to this previous work we provide a more mechanistic basis for the parameters in the model allowing us to derive limits on richness based on an environmental driver, temperature. A natural next step in this work would be to consider other properties of these equilibria such as their stability (capacity to resist perturbation) (*May, 1972*; *Allesina and Tang, 2012*; *Grilli et al., 2017*) or reactivity (the degree to which perturbations are amplified within the system) (*Neubert and Caswell, 1997*; *Arnoldi et al., 2018*). This would allow greater understanding of the dynamic behaviour of these systems across temperatures and allow us to identify whether and when microbial communities are more susceptible to disturbances at different points along thermal gradients. In this context, it is worth noting that feasible fixed points in the GLV are almost always stable (*Gibbs et al., 2018*), suggesting that patterns of stability-constrained richness should follow the same temperature response.

We also note that the GLV underlying our theory assumes a physically well-mixed system, that is, the spatial structure does not play a role. As such, spatial structure will impact species coexistence, for

instance, by localising competitive exclusion to spatial 'pockets.' We expect that future work incorporating spatial structure in our framework may reveal differences in the thermal responses of microbial species richness between environments with contrasting spatial structures (e.g. soil versus water).

Finally, we acknowledge that we have only considered competitive interactions here. Whilst it has been argued that competitive interactions dominate in microbial communities (*Foster and Bell, 2012*) there has more recently been a recognition of the importance of cooperative interactions that develop through cross-feeding between strains on their metabolic-by-products (*Goldford et al., 2018*; *Marsland et al., 2019*; *Lechón et al., 2021*). Though positive interactions can be considered in the GLV model framework this still represents an approximation of the resource dynamics that underlie cooperation in real communities (*Bunin, 2017*). Our approach towards determining the temperature dependence of trait distributions could, however, be applied to other models such as the recently-introduced microbial consumer-resource models (*Marsland et al., 2019*), which would allow explicit characterisation of resource-mediated interactions and thus the higher-order interactions and indirect effects that arise. We do not use this class of models here due to the additional complexity resource dynamics add and the existence of many analytical techniques to study the GLV. However, we would still expect the broad effects of distributions of thermal response parameters to have similar effects (as the thermal responses of traits are independent of the system dynamics) though the exact mapping of trait distributions (and the traits that need to be considered) on to richness may change.

Overall, our results provide a compelling theoretical basis and empirical evidence that the temperature-richness relationship in microbial communities can be strongly driven by variations in thermal physiology across species. Whilst often ignored, quantifying this variation in local communities are likely to be key to predicting the effects of temperature fluctuations on microbial community diversity across space and time.

## Methods
### Derivation of the theory

We begin with the GLV model of an *N*-species community where the biomass growth of the *i*th species given by

$$\frac{1}{x_i}\frac{dx_i}{dt} = r_i(T) - a_{ii}(T)x_i - \sum_{j \neq i}^{N} a_{ij}(T)x_j, \tag{1 revisited}$$

which is GLV_maintext in the main text. Here, $x_i$ is its biomass density (abundance) (mass · volume$^{-1}$), $r_i(T)$ it's intrinsic growth rate (time$^{-1}$), $a_{ij}(T)$ is the effect of interaction with the *j*th species' population (volume · mass$^{-1}$ · time$^{-1}$) (and thus $a_{ii}(T)$ is the strength of its intraspecific interactions).

### Mean-field approximation of the Lotka-Volterra Model

To determine the feasibility of a community in terms of the parameters in GLV_maintext and species richness, we need to first derive an expression for equilibrium biomass, $x_i^*$. Whilst it is possible to write GLV_maintext in matrix form and solve via inversion of the interaction matrix, this does not give a solution that is easily interpretable in terms of the parameters. As such we use a mean-field approximation which allows us to explicitly link the distributions of parameters to the equilibrium biomasses $x_i^*$ (*Wilson et al., 2003*; *Wilson and Lundberg, 2004*; *Rossberg, 2013*). By focusing on the averaged effect of interactions on each population's abundance, this approximation allows us to relate the equilibrium abundance vector to the mean pairwise interaction strengths $\langle a_{ij} \rangle$ across the community. We start by rewriting the summed interactions term for the *i*th species in the GLV model as:

$$\frac{\sum_{i \neq j}^{N} a_{ij} x_j}{N - 1} = \langle a_{ij} x \rangle,$$

$$\text{i.e., } \sum_{i \neq j}^{N} a_{ij} x_j = (N - 1) \langle a \rangle_{ij} \langle x \rangle + (N - 1)\text{Cov}(a_{ij}, x), \tag{7}$$

where the bar notation represents the average of the quantity across the $N - 1$ other species that the focal population can interact with (ignoring self-interaction). mean_int partitions the effects of interactions on the *i*th species' population into the average effect, $\langle a_{ij} \rangle \langle x \rangle$, and the covariance between strengths of the interactions and the heterospecifics' biomasses, $\text{cov}(a_{ij}, x)$. This mean-field

approximation assumes that system ($N$) is large, which ensures that the difference between the average heterospecific's biomasses and that of the focal species is small (as it is of order $N^{-1}$) and can thus be ignored. It also assumes that the second covariance term is negligible, which is equivalent to saying that any individual interaction between the focal species and another species' population has a small effect on its biomass abundance. Another way of framing this is that the variance in interaction strengths is not too large, a feature which can be seen by decomposing the covariance term into the correlation $\rho_{x,a_{ij}}$ and variance terms $\sigma_x$ and $\sigma_{a_{ij}}$

$$\text{cov}(a_{ij}, x) = \rho_{x,a_{ij}} \sigma_x^2 \sigma_{a_{ij}}^2 \tag{8}$$

Thus, the covariance term will be small as long as the correlation and the variation in interaction strengths are small.

Combining GLV_maintext and mean_int, we can express each species' population dynamics in terms the average interaction strength, giving the full mean-field model:

$$\frac{1}{x_i}\frac{dx_i}{dt} \approx r_i - a_{ii}x_i - (N-1)\langle a_{ij}\rangle \langle x\rangle. \tag{9}$$

Next, we obtain an expression for the community's dynamic equilibrium by setting *Equation 9* equal to zero and solving for $x_i$, giving:

$$x_i^* = \frac{r_i}{a_{ii}} - (N-1)\frac{\langle a_{ij}\rangle}{a_{ii}}\langle x\rangle^* \tag{10}$$

Then, taking the average across the $N$ populations and rearranging, the average biomass in the community is:

$$\langle x\rangle^* = \left\langle \frac{r}{a_{ii}}\right\rangle \frac{1}{1+(N-1)\langle a\rangle}.$$

Assuming that the growth rates and intraspecific interactions are independent i.e., $\text{cov}(r_i, a_{ii}) \approx 0$ we can write this as:

$$\langle x\rangle^* = \langle r\rangle \left\langle a_{ii}^{-1}\right\rangle \frac{1}{1+(N-1)\langle a\rangle}.$$

where $\left\langle a_{ii}^{-1}\right\rangle$ denotes the average inverse intraspecific interaction strength and $\langle a\rangle = \langle a_{ij}\rangle \left\langle a_{ii}^{-1}\right\rangle$ the product of the average of interspecific interaction and the inverse intraspecific interactions. By expressing interactions in this way the new term $\langle a\rangle$ measures the effective strength of competition in a community. This aligns with classic results from the ecological theory that species coexistence is based on the ratio of inter- and intraspecific competition. We can then substitute the expression for $\langle x\rangle$ into MF_partial to get equilibrium biomass:

$$x_i^* = \frac{r_i}{a_{ii}} - \frac{\langle r\rangle}{a_{ii}}\frac{(N-1)\langle a\rangle}{1+(N-1)\langle a\rangle}. \tag{11}$$

MF_equi shows how the equilibrium abundance reached by a population is a balance between its own growth and intraspecific interaction strength in the first term (which can be shown to be its carrying capacity by setting $a_{ij} = 0$ in GLV_maintext) minus the negative effect of interactions in the second. This second term includes both the average growth rate across the community as well as a saturating function of interactions. Biologically this makes sense because the effect of competition on a focal species' biomass depends on the abundance of its competitors in the environment (captured in the $\langle r\rangle$ term) and the strength of its interactions with them (captured by $(N-1)\langle a\rangle$). Because we assume interactions are competitive, they will always reduce population biomass relative to intrinsic carrying capacity.

## Condition for feasibility

Next, we use MF_equi to derive an expression for community feasibility—which sets the upper bound on species richness $N$—, in terms of species-level traits (i.e. the $r_i$'s and $a_{ij}$'s). A community is feasible if all its populations have non-zero equilibrium biomasses (i.e. $x_i^* > 0$) letting us write,

$$r_i' \quad > \quad \frac{(N-1)\langle a \rangle}{1+(N-1)\langle a \rangle} \quad \text{for all} \quad i = [1, 2, 3, \ldots, N] \tag{12}$$

Here, $r_i' = r_i/\langle r \rangle$ is the relative growth rate of the $i$th species (i.e. its value relative to the average across all $N$ populations). Feas_sp states that a community is feasible as long as the negative effects of competition on each population (RHS) do not outweigh its relative growth rate (LHS).

Using Feas_sp we next derive an expression for $P_{feas}$, the probability that a $N$-species community is feasible given the distribution of community-level trait values ($r_i'$'s and $a$'s). To do so we treat $r_i'$ and $a$ in Feas_sp as random variables that follow specific distributions (across species) in the community (denoted by the loss of subscript). This allows us to consider $r_i'$'s cumulative density function (CDF) which gives the probability that any given value of $r_i'$ is less than or equal to some value: $F_{r_i'}(x) = P(r_i' \leq x)$. Because the condition for feasibility states that $r_i'$ must be greater than the (negative) effect of interactions, we can use this CDF and the condition in Feas_sp to express $P_{feas}$ as

$$\begin{aligned} P_{feas} \quad &= P\left(r_i' > \frac{(N-1)\langle a \rangle}{1+(N-1)\langle a \rangle}\right)^N \\ &= \left[1 - F_{r_i'}\left(\frac{(N-1)\langle a \rangle}{1+(N-1)\langle a \rangle}\right)\right]^N, \end{aligned} \tag{2 revisited}$$

giving the probability of feasibility of an ecosystem as a function of species' traits. The expression is raised to the $N^{\text{th}}$ power because all $N$ populations within a community must themselves be feasible (the term in the brackets) for a system to be feasible.

## Incorporating thermal responses of traits

We now turn to the effect of temperature. First, we consider how the distribution of a given trait changes over temperature. We derive the distributions of the trait value in terms of the distributions of the thermal physiology parameters, which determine the shape of the thermal performance curve (TPC). We use the Boltzmann-Arrhenius equation to represent the temperature dependence of traits (**Gillooly et al., 2001**; **Savage, 2004**; **Dell et al., 2011**; **Dell et al., 2014**):

$$B(T) = B_0 e^{-\frac{E}{k}\left(\frac{1}{kT} - \frac{1}{kT_{ref}}\right)}. \tag{3 revisited}$$

Here, $B(T)$ is the trait value, $T$ is temperature in Kelvin, $B_0$ is the normalisation constant, i.e., the trait value at some reference temperature ($T_{ref}$, which we set to the middle of the OTR with no loss of generality, we can always obtain the same TPC for a given $T_{ref}$ by normalising $B_0$, $E$ (eV) is the thermal sensitivity which determines the change in trait value to a unit change of $1\,kT$, and $k$ is the Boltzmann constant. Although species-level thermal performance curves are generally unimodal, the Boltzmann-Arrhenius equation captures the rising portion (before the temperature of peak performance) of TPCs, which is also the temperature range within which populations typically operate (or experience) (the 'Operational Temperature Range,' or OTR; **Dell et al., 2011**; **Smith et al., 2019**; **Smith et al., 2021**). Indeed, the thermal optima of growth rates of mesophilic prokaryotes in laboratory experiments are typically 5–10°C higher than their (constant) ambient temperature **Smith et al., 2019**). Thus, focusing on the Boltzmann-Arrhenius portion of TPCs is relevant to the dynamics of real microbial communities, and also, conveniently, affords us analytic tractability.

We now consider how the TPC parameters $B_0$ and $E$ of growth ($r_i$'s) and interaction rates ($a_{ij}$'s) vary across species within the community and how this variation is propagated through Boltzmann to give the community-level distributions of these two traits at different temperatures. We begin with the natural log of Boltzmann:

$$\log(B(T)) = \log(B_0) - \frac{E}{k}\left(\frac{1}{kT} - \frac{1}{kT_{ref}}\right). \tag{13}$$

Next, we assume that $\log(B_0)$ and $E$ are distributed as a multivariate normal distribution such that:

$$\begin{bmatrix} \log(B_0) \\ E \end{bmatrix} \sim \mathcal{N}\left(\begin{bmatrix} \mu_{B_0} \\ \mu_E \end{bmatrix}, \begin{bmatrix} \sigma_{B_0}^2 & \sigma_{B_0,E} \\ \sigma_{B_0,E} & \sigma_E^2 \end{bmatrix}\right)$$

where $\mu_{B_0}$ and $\mu_E$ are the respective means and $\sigma_{B_0}^2$ and $\sigma_E^2$ the variances of the normalisation constant and thermal sensitivity, respectively, and $\sigma_{B_0,E}$ is the covariance between them. $B_0$ is indeed expected to be log-normally distributed for growth and interaction rates (*Kontopoulos et al., 2020*; *Dell et al., 2014*; *Bestion et al., 2018*). On the other hand, $E$ distributions tend to be right-skewed (*Kontopoulos et al., 2020*; *Smith et al., 2019*; *Dell et al., 2011*), but we use the normal distribution here as an adequate approximation. Then, because LogBoltzmann is a linear combination of two co-varying normally-distributed random variables, $\log(B(T))$ will itself be normally distributed as

$$\log(B(T)) \sim \mathcal{N}\left(\mu_B(T), \sigma_B^2(T)\right) \quad \text{where} \quad \begin{aligned} \mu_B(T) &= \mu_{B_0} - \mu_E\left(\frac{1}{kT} - \frac{1}{kT_{ref}}\right) \\ \sigma_B^2(T) &= \sigma_{B_0}^2 + \sigma_E^2\left(\frac{1}{kT} - \frac{1}{kT_{ref}}\right)^2 - 2\sigma_{B_0,E}\left(\frac{1}{kT} - \frac{1}{kT_{ref}}\right). \end{aligned} \tag{4 revisited}$$

That is, the temperature-specific trait values across species in a community for either growth or interaction rate can be represented by a log-normal distribution. boltz_maintext shows how:

1. The mean trait value across species at a given temperature ($\mu_B(T)$) increases with the mean baseline trait value $\mu_{B_0}$ s as well as the mean thermal sensitivity $\mu_E$ s. Note that $-\mu_E$ still implies a positive gradient with respect to temperature because we are dealing with inverse temperature ($1/kT$).
2. Variation in the trait's value across species ($\sigma_{B_0}^2$) increases with the variance in the baseline trait value $\sigma_{B_0}^2$.
3. Trait variation decreases to a minimum at some intermediate temperature because the quadratic term $\sigma_E^2\left(\frac{1}{kT} - \frac{1}{kT_{ref}}\right)^2$ is convex (concave upward) due to the inverse temperature scale.
4. The temperature at which this minimum trait variation occurs is modulated by the covariance term ($2\sigma_{B_0,E}\left(\frac{1}{kT} - \frac{1}{kT_{ref}}\right)$). A negative covariance between the two TPC parameters will increase the temperature of minimum trait variance while a positive covariance will decrease it.

The temperature of the lowest trait variation determined by boltz_maintext is key because it determines the location of the peak of the temperature-richness relationship, as we will show below. Henceforth, we choose $T_{ref}$ to always be the center of the OTR (~13°C based on our empirical data synthesis; see below). Note that our results are qualitatively independent of our choice of $T_{ref}$ as one can always recover the same trait distribution by altering the variance $\sigma_{B_0}^2$ and covariance $\sigma_{B_0,E}^2$ terms.

It is useful to consider the exact conditions under which the variance in a trait is sufficient to cause unimodal responses. Using the definition for the average of a log-normal distribution $m = \exp(\mu + \sigma^2/2)$ and substituting the expressions in boltz_maintext we obtain

$$m = \exp\left(\frac{\sigma_E^2 \Delta_T^2}{2} - (\mu_E + \sigma_{B_0,E})\Delta_T + \mu_{B_0} + \frac{\sigma_{B_0}^2}{2}\right) \tag{14}$$

where $\Delta_T = \left(1/kT_{ref} - 1/kT\right)$. To consider the unimodality we can then consider the point at which the square term above dominates. For ecologically relevant temperatures (0–40°C) and a reference temperature at 20° the value of $\Delta_T$ will vary from ~2.9–2.5 so we can consider the case when $|\Delta_T| = 2$ giving the condition

$$\sigma_E^2 > \mu_E + \sigma_{B_0,E}. \tag{15}$$

This shows a lower bound amount of variation in thermal sensitivity to observe unimodal responses. The degree of variation must be greater than the average thermal sensitivity plus any covariance.

Note that as the covariance is expected to be negative, increasing covariance increases the unimodality of the thermal response.

## Temperature dependence of species richness

Next, we use (boltz_maintext) to derive the distribution of $r_i'$ as well as the value of $\langle a \rangle$, which together determine feasibility (*Equation 2*; *Figure 1D*). First, recall that:

$$r_i'(T) = \frac{r_i(T)}{\langle r \rangle (T)}. \tag{16}$$

Then, because $r_i(T)$'s TPC follows a Boltzmann-Arrhenius relationship, its TPC parameters are distributed as in boltz_maintext and its mean (as a log-normally distributed variable) is given as:

$$\langle r \rangle (T) = e^{\mu_r(T) + \frac{\sigma_r(T)^2}{2}}.$$

Substituting this into *Equation 16* and taking the natural log gives:

$$\log(r_i'(T)) = \log(r_i(T)) - \mu_r(T) - \frac{\sigma_r(T)^2}{2}.$$

as $\log(r')(T)$ is normally distributed this represents a simple shift in its mean giving,

$$\log(r_i(T)) \sim \mathcal{N}\left(-\frac{\sigma_r(T)^2}{2}, \sigma_r(T)\right). \tag{5 revisited}$$

Next consider the thermal dependence of $\langle a \rangle$ which depends on the interaction strength distributions $a_{ij}(T)$ and $a_{ii}(T)$. Because the interactions are also assumed to follow a Boltzmann-Arrhenius response, their distributions are also log-normally distributed as in boltz_maintext. We can, therefore, obtain its average with the expression

$$
\begin{aligned}
\langle a \rangle (T) &= \langle a_{ij} \rangle (T) \left\langle a_{ii}^{-} \right\rangle (T) \\
&= \left[\exp\left(\mu_{a_{ij}}(T) + \frac{\sigma_{a_{ij}}(T)^2}{2}\right)\right] \left[\exp\left(-\mu_{a_{ii}}(T) + \frac{\sigma_{a_{ii}}(T)^2}{2}\right)\right]. \\
&= \exp\left(\mu_{a_{ij}}(T) - \mu_{a_{ii}}(T) + \frac{\sigma_{a_{ij}}(T)^2 + \sigma_{a_{ii}}(T)^2}{2}\right)
\end{aligned}
\tag{6 revisited}
$$

Note the negative sign of the average intraspecific interaction strength which arises as we consider the mean of the inverse of $a_i i$. The two equations, *Equations 5 and 6*, show how the thermal responses of $r_i'$ and $\langle a \rangle$ are both driven by the variance in the underlying log-trait distribution (and thus the variance in thermal sensitivity $\sigma_E^2$ and covariance $\sigma_{B_0,E}$) with $\langle a \rangle$ additionally being driven by the average log-trait value (and therefore, its average thermal sensitivity, $\mu_{E,a}$). The effects of this on richness are detailed in the main text.

## Empirical data

In order to obtain empirically relevant estimates of the mean, variance, and covariance of $B_0$ and $E$ we used data from both *Smith et al., 2021* who experimentally measured the thermal performance (growth rate) of 29 strains of environmentally isolated bacteria and *Smith et al., 2019* who synthesised data from existing bacterial thermal performance experiments for 422 stains. For both datasets, took the original data and fit the Sharpe Schoolfield model which describes the unimodal thermal response of traits to temperature (including $B_0$ and $E$ values) using the rTPC package (*Schoolfield et al., 1981*; *Padfield et al., 2021*). We rejected any fits that had non-significant ($p < 0.05$) parameter estimates or did not converge. Taking the fitted $B_0$ and $E$ values, normalised the $B_0$ values by dividing by the mean to allow comparison across the datasets, and filtered out the values of $\log(B_0)$ larger than −15. We then fitted the multivariate-normal distribution using maximum likelihood estimation (MLE; *Besançon et al., 2021*) giving estimates for the means and variance-covariance matrix, which can be used to generate temperature-dependent distributions of growth rate across the community

boltz_maintext. We used these parameters to estimate temperature-richness relationships using the method described in the previous section with both $r$ and $a$ TPC parameters set to the same values except for the $\mu_{B_0}$ values which were set to 0.0 and –5.0 for $\log(r_0)$ and $\log(a_0)$, respectively.

## Acknowledgements

We thank Tom Smith for helping with the empirical datasets and Pankaj Mehta and Tom Bell for discussions and feedback on an earlier version of the manuscript.

## Additional information

### Funding

| Funder | Grant reference number | Author |
| --- | --- | --- |
| Natural Environment Research Council | NERC QMEE Centre for Doctoral Training NE/P012345/1 | Tom Clegg Samraat Pawar |
| Leverhulme Trust | RF-2020-653\2 | Samraat Pawar |
| NERC Natural Environment Research Council | NE/M020843/1 | Samraat Pawar |
| NERC Natural Environment Research Council | NE/S000348/1 | Tom Clegg Samraat Pawar |

The funders had no role in study design, data collection and interpretation, or the decision to submit the work for publication.

### Author contributions

Tom Clegg, Conceptualization, Formal analysis, Writing - original draft, Writing - review and editing; Samraat Pawar, Conceptualization, Supervision, Writing - original draft, Writing - review and editing

### Author ORCIDs

Tom Clegg ⓘ https://orcid.org/0000-0002-8381-3132

### Decision letter and Author response

Decision letter https://doi.org/10.7554/eLife.84662.sa1
Author response https://doi.org/10.7554/eLife.84662.sa2

## Additional files

### Supplementary files
• MDAR checklist

### Data availability

The current manuscript is a computational study, so no data have been generated for this manuscript. Modelling code is available on GitHub (copy archived at *Clegg, 2024*).

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
