## [Editor Report]

This important study proposes a phenomenologically motivated theoretical framework to explain observed patterns of the temperature dependence of microbial diversity. The methodology is overall convincing. The manuscript should be of interest to microbial ecologists.

---

## [Decision Letter]

**Decision letter after peer review:**

Thank you for submitting your article "Variation in thermal physiology can drive the temperature-dependence of microbial community richness" for consideration by *eLife*. Your article has been reviewed by 3 peer reviewers, and the evaluation has been overseen by a Reviewing Editor and Aleksandra Walczak as the Senior Editor. The reviewers have opted to remain anonymous.

Essential revisions (for the authors):

1) Please clarify, and spell out in detail, what assumptions and approximations are made in calculations, and at what stages. In the presentation of the mean-field approximation, it should be explicitly said that extra have been made approximations (e.g. when computing the mean of the inverse). Moreover, given these approximations, it is important to clarify when they break down and when they work well, both with explanations and concrete examples.

2) Please clarify the manuscript. In particular, please clearly define each notation and motivate parameter or function choices. Please address all of the reviewers' points to improve this.

3) Please discuss why a consumer-resource model was not chosen and what it might change. Please also discuss the motivation of the temperature dependence of the interaction parameters.

*Reviewer #1 (Recommendations for the authors):*

1) The derivation of the mean field result is not correct. It may hold in some specific conditions that are not properly discussed. The problem is after Eq. (8) of the Methods section. Tacking the average across the N populations does not lead to the following equation for the average stationary population. In fact, denoting by 〈∙〉 the average, then the mean field consistency equation should read: x∗¯=⟨riaii⟩+(N−1)aij¯⟨1aii⟩x∗¯ but ⟨riaii⟩ ≠ ⟨ri⟩⟨aii⟩ and ⟨1aii⟩ ≠ 1⟨aii⟩. Therefore, the mean field result presented in the paper, in general are not correct. In some specific cases, e.g. the random variables a_ii_ is sharply peaked around its mean, then it may hold that ⟨riaii⟩ ≈ ⟨ri⟩⟨aii⟩ and ⟨1aii⟩ ≈ 1⟨aii⟩.

Decision letter images 1, 2 and 3 show numerically the comparison of ⟨riaii⟩ with ⟨ri⟩⟨aii⟩ 〈_a_*^r^*ii^i^〉 with _〈_^〈^_a_*^r^*ii^i〉^_〉_ and of ⟨1aii⟩ with 1⟨aii⟩ for three different cases (mean and variances highlighted as plot label):

**Decision letter image 1. sa1fig1:** ri and aii drawn from a Log Normal Distribution with “high” variance.

**Decision letter image 2. sa1fig2:** ri and aii drawn from a Log Normal Distribution with “low” variance.

**Decision letter image 3. sa1fig3:** ri and aii drawn from a Log Normal Distribution with “low” variance.

So it is clear that actually, in the log-normal case, that should be the actual distribution from where r_i_ and a_ii_ have been drawn, the average of the ratio of the two random variables cannot be substituted with the ratio of the averages.

Relatedly, given the above results, it is not clear to me, how it is possible that the approximation proposed by the authors work so well, for example in Figure 2. Moreover, it is not clear to me how r_i_, a_ij_ and a_ii_ are chosen in the numerical simulation of the full GLV. Are they drawn from a LogNormal distribution? Just after Eq. 2 it seems that indeed they are lognormal distributed, but this should be specified better in the Figures and also it is necessary adding information about which parameters have been used. Moreover, how much the goodness of the analytical approximation depends on the specific choices of the parameters? I think that a sensitivity analysis and related discussion on the limitation of the analytical approximations are needed.

In general, I think that it should have been more appropriate to perform a more advanced mean field approximation, for example following the work “Collapse of resilience patterns in generalized Lotka-Volterra dynamics and beyond” (Tu, C., Grilli, J., Schuessler, F., and Suweis, S. (2017). Physical Review E, 95(6), 062307), from which a similar approximation of the effective average population could be derived. Moreover, using this approximation, it is possible to go beyond purely competitive ecosystems, as it holds also for communities with mutualistic interactions. In fact, the statement that GLV only works for competitive communities is not correct (there are many works using GLV with (also) positive interactions (e.g. Rohr, Rudolf P., Serguei Saavedra, and Jordi Bascompte). "On the structural stability of mutualistic systems." Science 345.6195 (2014): 1253497; Suweis, S., Simini, F., Banavar, J. R., and Maritan, A. (2013). Emergence of structural and dynamical properties of ecological mutualistic networks. Nature, 500(7463), 449-452.).

While it is quite clear the physiological dependence of the growth rate on temperature, it is not quite evident why also the interactions strengths should depend on the interactions strengths a_ij._ How the works conclusions would change if only r_i_ depends on time (see also Abreu, C. I., Dal Bello, M., Bunse, C., Pinhassi, J., and Gore, J. (2022). *Warmer temperatures favor slower-growing bacteria in natural marine communities*. bioRxiv).The section *The theory holds in dynamically-assembled communities* is hard to read, as it lacks of the definition of what is a dynamically-assembled community, how it is mathematically defined and why you also want to explore such a case. Some information must be available in the main text, some other you can refer (but please explicitly put the link) to the Methods section.

*Reviewer #2 (Recommendations for the authors):*

In their paper *Variation in thermal physiology can drive the temperature dependence of microbial community richness*, Clegg and Parwar present a relatively simple phenomenological model for explaining the wide variety of empirically observed relationships between temperature and diversity in the microbial world. Previous theories such as the Metabolic theory of biodiversity (MTB) and the metabolic niche hypothesis have emphasized the role of energy through either more efficient cellular kinetics or temperature dependent niches. This paper builds on these works by showing that if one accounts for variation of temperature sensitivity across species, one can get a much richer set of behaviors consistent with empirical observations.

Overall, I find the manuscript quite compelling and the model presented as a very nice summary of how variability in temperature dependence, simple Arrhenius scaling, and arguments based on modern coexistence theory can be combined to explain empirical observations of species abundance distributions and temperature. I find Figures 2 and 3 quite interesting and they have the virtue of resolving a major puzzle in the current literature and proposing concrete mechanistic hypothesis. For all these reasons, I think this manuscript makes an important contribution to the literature and I recommend publishing in *eLife*.

However, I do have some comments and concerns that I think would be helpful for the authors to address.

I feel like the manuscript is too terse and hard to follow. For example, the parameter *E* is not defined explicitly anywhere in the main text. I would suggest that the incorporating thermal responses of traits (including Equations 13 and 14) be moved to the main text and this discussion greatly expanded. I could not follow what was going on.

I do not understand the physical/ecological motivation for log*B*_0_ and *E* are anti-correlated. Does this follow from theory or empirical fits? How do we know that the experiments from Smith et al. hold more generally?

Currently, parameters are drawn from a log-normal distribution. This means that it is long-tailed. Do the general trends they hold for non-long tailed distributions. I understand that the *r*_i_ must be positive, but this can be done by for example, using a truncated Gaussian. If the long tails are essential, could you please explain why the tails matter? The form of ¯*r* below Eq. 15 would suggest that the results here may depend very strongly on the long tailed distribution assumption. It would be nice to understand how the phenomenology changes if this is not the case.

I feel like the averages below equation 7 are done sloppily. The agreements with numerics suggest these are small effects but in reality we have that

x∗=1N∑ixi∗=1N∑i(riaii−(N−1)1N)∑ia¯IIaiix¯∗ (1)

Notice that in general that

1N∑i(riaii≠r¯a¯ii,) (2)

since this is not how averages work. A similar thing holds for the second term. For this reason, the expressions are not correct even under the MFT assumption. Numerics suggest this does not matter but this approximation should be made more clear.

The caption for Figure 4 seems to be cut-off.

Can the author please discuss why they think the predictions of Figures 3b,c fail in greater detail?

*Reviewer #3 (Recommendations for the authors):*

I think the paper does make inroads into an important question, and the focus on the temperature-dependence of species interactions does go beyond what has been assumed e.g. in metabolic theory. My main questions are detailed in the public review. Namely:

- To what extent is the mean-field approximation for x* (which I think can be interpreted as an approximation for the inverse of a matrix with entries a_ij) valid for the full range of values of a_ij.

There is also a long literature on feasibility analyses, going back at least to the `70s (e.g. Goh and Jennings, 1977, Ecological Modeling), and some of this is pertinent e.g. in relating to the authors' results for the probability of feasibility and how this depends on the number of species present. It would be helpful to engage with this literature.

- In general, I do not fully understand the justification for the functional forms of growth rate and interaction rate on temperature. The latter (the way the a_ij are assumed to depend on the temperature) seems particularly difficult to pin down. Is there any clear justification for this form?

- Whatever the temperature dependence, in Lotka-Volterra it seems inevitable that this will be a phenomenological assumption. The way the authors build up in the introduction, I thought they were headed towards a consumer-resource model, maybe even with intracellular dynamics determining the temperature dependence of interactions. This would be the approach e.g. of the Droop model (also going back to the 70s), or e.g. work from a couple of years ago (Muscarella and O'Dwyer, 2020). I am not claiming that we can't drop explicit resources, reduce to Lotka-Volterra, and make progress. But it makes it hard to understand how robust the results are to the authors' assumptions about the way Lotka-Volterra parameters change with environmental context.

---

## [Author Response]

Essential revisions (for the authors):1) Please clarify, and spell out in detail, what assumptions and approximations are made in calculations, and at what stages. In the presentation of the mean-field approximation, it should be explicitly said that extra have been made approximations (e.g. when computing the mean of the inverse). Moreover, given these approximations, it is important to clarify when they break down and when they work well, both with explanations and concrete examples.

We have made extensive revisions to the Results section, significantly increasing the detail (and hopefully clarity) of the text. This includes stating the assumptions more explicitly when they are made. We have also corrected an error in the derivation of the mean field approximation, removing the need to make assumptions about the mean of the inverse. We have also included a sensitivity analysis to make clear the conditions under which the approximation does not work.

2) Please clarify the manuscript. In particular, please clearly define each notation and motivate parameter or function choices. Please address all of the reviewers' points to improve this.

As noted above we have rewritten most of the results and methods to make the derivation of the theory clearer. This includes explicitly defining parameters and the assumptions in the main text.

3) Please discuss why a consumer-resource model was not chosen and what it might change. Please also discuss the motivation of the temperature dependence of the interaction parameters.

We have included explicit discussion of the pros and cons of using a consumer-resource type model in lines 337-346. In brief we chose the GLV because of the analytic tractability and existence of methods such as the mean field approximation for its analysis.

Though using a consumer-resource model would change the exact results, we expect the general effects of thermal physiology parameters (i.e variation and covariance determining the magnitude and location of peak richness) would remain and thus some of the broad results would be the same. This ultimately is a promising avenue for future research and we have included it explicitly in the discussion.

Reviewer #1 (Recommendations for the authors):1) The derivation of the mean field result is not correct. It may hold in some specific conditions that are not properly discussed. The problem is after Eq. (8) of the Methods section. Tacking the average across the N populations does not lead to the following equation for the average stationary population. In fact, denoting by 〈∙〉 the average, then the mean field consistency equation should read:

x∗¯=⟨riaii⟩+(N−1)aij¯⟨1aii⟩x∗¯

but ⟨riaii⟩ ≠ ⟨ri⟩⟨aii⟩ and ⟨1aii⟩ ≠ 1⟨aii⟩. Therefore, the mean field result presented in the paper, in general are not correct. In some specific cases, e.g. the random variables a_ii_ is sharply peaked around its mean, then it may hold that ⟨riaii⟩ ≈ ⟨ri⟩⟨aii⟩ and ⟨1aii⟩ ≈ 1⟨aii⟩.Decision letter images 1, 2 and 3 show numerically the comparison of ⟨riaii⟩ with ⟨ri⟩⟨aii⟩ and of ⟨1aii⟩ with 1⟨aii⟩ for three different cases (mean and variances highlighted as plot label).So it is clear that actually, in the log-normal case, that should be the actual distribution from where r_i_ and a_ii_ have been drawn, the average of the ratio of the two random variables cannot be substituted with the ratio of the averages.

We thank the reviewer for pointing out this error in the derivation of the mean field.

We have amended the derivation which requires two new steps (lines 398-407):

(a) Specifically accounting for the mean of the reciprocal of intraspecific interactions ⟨aii−1⟩. This means the effective interaction strength is redefined as ⟨a⟩=⟨aij⟩⟨aii−1⟩. Note that the inverse of a log-normal variable still has a log-normal distribution allowing us to write an expression for the temperature dependence of the average.

(b) Assuming that growth rates and intraspecific interactions are independent cov(ri,aii)≈0. This allows the average of the product to be written as: ⟨raii⟩=⟨r⟩⟨aii−1⟩ with these two changes the derivation is more explicit in its assumptions. Whilst the structure of the equations changes slightly this does not affect the overall effect of the parameters on the behavior of the model. We have generated a new set of results with this updated model including the numerical simulations.

Relatedly, given the above results, it is not clear to me, how it is possible that the approximation proposed by the authors work so well, for example in Figure 2. Moreover, it is not clear to me how r_i_, a_ij_ and a_ii_ are chosen in the numerical simulation of the full GLV. Are they drawn from a LogNormal distribution? Just after Eq. 2 it seems that indeed they are lognormal distributed, but this should be specified better in the Figures and also it is necessary adding information about which parameters have been used. Moreover, how much the goodness of the analytical approximation depends on the specific choices of the parameters? I think that a sensitivity analysis and related discussion on the limitation of the analytical approximations are needed.

We have added a lot more detail on the methods in the main text which hopefully make the procedure for simulating and predicting patterns of richness across temperature more clear (lines 214-230). We have also added explicit discussion of the limitations of the assumptions in the model in lines (see response to comment 1) as well as a new sensitivity analysis (see lines 232-245).

In general, I think that it should have been more appropriate to perform a more advanced mean field approximation, for example following the work “Collapse of resilience patterns in generalized Lotka-Volterra dynamics and beyond” (Tu, C., Grilli, J., Schuessler, F., and Suweis, S. (2017). Physical Review E, 95(6), 062307), from which a similar approximation of the effective average population could be derived. Moreover, using this approximation, it is possible to go beyond purely competitive ecosystems, as it holds also for communities with mutualistic interactions. In fact, the statement that GLV only works for competitive communities is not correct (there are many works using GLV with (also) positive interactions (e.g. Rohr, Rudolf P., Serguei Saavedra, and Jordi Bascompte. "On the structural stability of mutualistic systems." Science 345.6195 (2014): 1253497; Suweis, S., Simini, F., Banavar, J. R., and Maritan, A. (2013). Emergence of structural and dynamical properties of ecological mutualistic networks. Nature, 500(7463), 449-452).).

Whilst it is of course always possible to use a more complex approximation we have chosen to use the simplest form of the mean field approximation (sometimes called a “naive mean field”). We argue that this approximation is adequate to demonstrate the effect of variation in the thermal physiology parameters as a mechanism for community level richness.

In the paper mentioned Tu et al. (2017) demonstrate the conditions under which a similar mean field type approximation works for Lotka-Volterra systems. They show that the approximation becomes less accurate as variation in interaction rates increase. This result makes sense and is related to the assumption that interactions are relatively uncorrelated with abundances in the mean field derivation. It is easy to show that all else being equal, increasing variation in interactions increases the magnitude of the covariance term, breaking the assumptions of the mean field approximation. This assumption and the link to variation in interactions is explicitly discussed in lines 382-393.

We have also amended the language with respect to the cooperative interactions, we acknowledge that they can be considered in the GLV model and have altered the text (lines 346-360).

2)While it is quite clear the physiological dependence of the growth rate on temperature, it is not quite evident why also the interactions strengths should depend on the interactions strengths a_ij._ How the works conclusions would change if only r_i_ depends on time (see also Abreu, C. I., Dal Bello, M., Bunse, C., Pinhassi, J., and Gore, J. (2022). Warmer temperatures favor slower-growing bacteria in natural marine communities. bioRxiv).

We argue that exponential-like temperature dependence of interactions is likely due to their dependence on metabolic rates of interacting populations. In the case of microbes the strength of competition will primarily be determined by uptake rates which are known to depend on metabolism and thus be described by biochemical kinetics. Indeed expressions for the strength of interactions derived from more mechanistic consumer resource models such as Marsland et al. (2020; Scientific Reports, 10) demonstrate this dependence well. We have updated the manuscript in lines 147-159 to include this reasoning.

3) The section The theory holds in dynamically-assembled communities is hard to read, as it lacks of the definition of what is a dynamically-assembled community, how it is mathematically defined and why you also want to explore such a case. Some information must be available in the main text, some other you can refer (but please explicitly put the link) to the Methods section.

We agree with the reviewer here, the section was unclear and added little to the readers understanding of the main focus of the effects of variation in thermal physiology. As such we have chosen to remove this section of the paper. The topic of community assembly with immigration-extinction dynamics adds a lot of additional complexity and the mean field approximation we apply does not extend well to such situations.

Reviewer #2 (Recommendations for the authors):In their paper Variation in thermal physiology can drive the temperature dependence of microbial community richness, Clegg and Parwar present a relatively simple phenomenological model for explaining the wide variety of empirically observed relationships between temperature and diversity in the microbial world. Previous theories such as the Metabolic theory of biodiversity (MTB) and the metabolic niche hypothesis have emphasized the role of energy through either more efficient cellular kinetics or temperature dependent niches. This paper builds on these works by showing that if one accounts for variation of temperature sensitivity across species, one can get a much richer set of behaviors consistent with empirical observations.Overall, I find the manuscript quite compelling and the model presented as a very nice summary of how variability in temperature dependence, simple Arrhenius scaling, and arguments based on modern coexistence theory can be combined to explain empirical observations of species abundance distributions and temperature. I find Figures 2 and 3 quite interesting and they have the virtue of resolving a major puzzle in the current literature and proposing concrete mechanistic hypothesis. For all these reasons, I think this manuscript makes an important contribution to the literature and I recommend publishing in eLife.However, I do have some comments and concerns that I think would be helpful for the authors to address.I feel like the manuscript is too terse and hard to follow. For example, the parameter E is not defined explicitly anywhere in the main text. I would suggest that the incorporating thermal responses of traits (including Equations 13 and 14) be moved to the main text and this discussion greatly expanded. I could not follow what was going on.

We agree with the reviewer that the clarity of the manuscript needed improvement. As noted in our reply to comment 4 we have made significant changes (such as including additional equations in the main text) which we hope make the manuscript flow better.

I do not understand the physical/ecological motivation for logB_0_ and E are anti-correlated. Does this follow from theory or empirical fits? How do we know that the experiments from Smith et al. hold more generally?

We have now clarified this on lines 164-179 of the manuscript. In short, the anticorrelation arises from the trade-off between thermal specialists (species that perform high at a narrow temperature rang have high sensitivity E and low baseline performance B0) and generalists (species perform at a lower level but across a wider range high performance B0 and low sensitivity E). This trade off arises as species cannot perform highly across all temperatures and must either invest in either strategy.

Currently, parameters are drawn from a log-normal distribution. This means that it is long-tailed. Do the general trends they hold for non-long tailed distributions. I understand that the r_i_ must be positive, but this can be done by for example, using a truncated Gaussian. If the long tails are essential, could you please explain why the tails matter? The form of ¯r below Eq. 15 would suggest that the results here may depend very strongly on the long tailed distribution assumption. It would be nice to understand how the phenomenology changes if this is not the case.

In general long-tailed distributions are not required and the relationship between the parameter distributions and richness are not affected. The only way the specific shape of the distribution would matter is if long tails resulted in the violation of the small covariance assumption, that long tails in interaction strengths meant that some individual interactions had a disproportionate effect on biomass. This is a limitation of the mean field approximation and is clearly discussed in the main text (lines 387-393). It is important to note that the specific distribution of the traits arises mechanistically from the exponential-like temperature dependence at the individual population level. Given this we are confident in the appropriateness of the distribution describe community-level trait distributions.

I feel like the averages below equation 7 are done sloppily. The agreements with numerics suggest these are small effects but in reality we have that (The remaining comment is qualitatively identical to Reviewer 1, comment 1)

Please see our response to Reviewer 1, comment 1.

Reviewer #3 (Recommendations for the authors):I think the paper does make inroads into an important question, and the focus on the temperature-dependence of species interactions does go beyond what has been assumed e.g. in metabolic theory. My main questions are detailed in the public review. Namely:- To what extent is the mean-field approximation for x* (which I think can be interpreted as an approximation for the inverse of a matrix with entries a_ij) valid for the full range of values of a_ij_.

We thank the reviewer for pointing this out and we agree that the language used is imprecise. The GLV model is solvable using the matrix inversion method but as they note, this does not give an interpretable expression in terms of the system parameters. This is important as we aim to build understanding of how these parameters (which in turn depend on temperature) affect the richness in communities. We have made this clearer in lines 372-379.

In regards to the validity of the approximation we have significantly increased the detail of the method in the manuscript, including the assumptions it makes (lines 384-393). In general the method assumes that any individual interaction has a weak effect on abundance. This will fail when the variation in interactions becomes too strong but should be robust to changes in the average interaction strength across the community.

There is also a long literature on feasibility analyses, going back at least to the `70s (e.g. Goh and Jennings, 1977, Ecological Modeling), and some of this is pertinent e.g. in relating to the authors' results for the probability of feasibility and how this depends on the number of species present. It would be helpful to engage with this literature.

We thank the reviewer and have added the reference to the main text (lines 323-327)

- In general, I do not fully understand the justification for the functional forms of growth rate and interaction rate on temperature. The latter (the way the a_ij are assumed to depend on the temperature) seems particularly difficult to pin down. Is there any clear justification for this form?

We thank the reviewer for their comment. The assumption of positive growth rates is indeed a feature of the Boltzmann-Arrhenius model of temperature dependence. We use the Boltzmann-Arrhenius model due to the dependence of growth on metabolic rate. As metabolic rate is ultimately determined by biochemical kinetics its temperature dependence is well described by the Boltzmann-Arrhenius. In addition to this reasoning there is a wealth of empirical evidence supporting the use of the Boltzmann-Arrhenius to describe the temperature dependence of growth rate in microbes.

Ultimately the temperature dependence of resource supply is not something we can directly consider in our model. As such we have to assume that resource supply is sufficient to maintain positive growth rates in the community. Note that this assumption only requires resource supply is sufficient to maintain positive growth rates (i.e. the maximal growth rate of species in isolation) not that resource supply is sufficient to maintain growth in the presence of intra- and interspecific competition. We have updated the manuscript in lines 156-159 to make these assumptions more clear.

The reviewer is correct, it is very difficult to measure interaction coefficients experimentally and to our knowledge there is little to no data available on their empirical temperature responses. We as a best guess use the observed variation in thermal physiology parameters for growth rate as a proxy assuming that interactions must also depend on metabolic rates of the interacting species.

See also response to Reviewer 1, comment 2.

- Whatever the temperature dependence, in Lotka-Volterra it seems inevitable that this will be a phenomenological assumption. The way the authors build up in the introduction, I thought they were headed towards a consumer-resource model, maybe even with intracellular dynamics determining the temperature dependence of interactions. This would be the approach e.g. of the Droop model (also going back to the 70s), or e.g. work from a couple of years ago (Muscarella and O'Dwyer, 2020). I am not claiming that we can't drop explicit resources, reduce to Lotka-Volterra, and make progress. But it makes it hard to understand how robust the results are to the authors' assumptions about the way Lotka-Volterra parameters change with environmental context.

See also response to Essential revisions, comment 3.